# Immunoregulatory properties of erythroid nucleated cells induced from CD34+ progenitors from bone marrow

**Julia A. Shevchenko**[�], **Roman Yu Perik-Zavodskii**[�], **Kirill V. Nazarov**[iD][‡], **Vera V. Denisova**[�], **Olga Yu. Perik-Zavodskaya**[�], **Yulia G. Philippova**[‡], **Alaa Alsalloum**[‡], **Sergey V. Sennikov**[iD]*[�he]

Laboratory of Molecular Immunology, Federal State Budgetary Scientific Institution "Research Institute of Fundamental and Clinical Immunology", Novosibirsk, Russia

he These authors contributed equally to this work.
‡ KVN, YGP and AA also contributed equally to this work.
* sennikovsv@gmail.com

**Data Availability Statement:** All relevant data are within the Paper. In the ZENODO repository, Yulia Shevchenko posted minimal data on cytometric

## Abstract

CD 71+ erythroid nucleated cells have pronounced immunoregulatory properties in normal and pathological conditions. Many populations of cells with immunoregulatory properties are considered candidates for cellular immunotherapy for various pathologies. This study characterized the immunoregulatory properties of CD71+ erythroid cells derived from CD34-positive bone marrow cells under the influence of growth factors that stimulate differentiation into erythroid cells. CD34-negative bone marrow cells were used to isolate CD71+ erythroid nuclear cells. The resulting cells were used to assess the phenotype, determine the mRNA spectrum of the genes responsible for the main pathways and processes of the immune response, and obtain culture supernatants for the analysis of immunoregulatory factors. It was found that CD71+ erythroid cells derived from CD34+ cells carry the main markers of erythroid cells, but differ markedly from natural bone marrow CD71+ erythroid cells. The main differences are in the presence of the CD45+ subpopulation, distribution of terminal differentiation stages, transcriptional profile, secretion of certain cytokines, and immunosuppressive activity. The properties of induced CD71+ erythroid cells are closer to the cells of extramedullary erythropoiesis foci than to natural bone marrow CD71+ erythroid cells. Thus, when cultivating CD71+ erythroid cells for clinical experimental studies, it is necessary to take into account their pronounced immunoregulatory activity.

## Introduction

Erythropoiesis is a tightly regulated multi-stage process that originates from a multipotent stem cell and ends with a mature enucleated erythrocyte [1]. Human erythropoiesis includes the stages of early erythropoiesis, terminal erythroid differentiation, and maturation of reticulocytes [2]. The terminal erythroid differentiation phase is subdivided chronologically into four stages, proerythroblast (ProE), basophilic erythroblast (BasoE), polychromatophilic erythroblast (PolyE), and orthochromatic erythroblast (OrthoE), based on the morphological characteristics of the cells [3]. The main function of erythroid nucleated cells is to maintain a pool

analysis of cells, cytokine production, and the results of a mixed leukocyte reaction. https://zenodo.org/record/7865730. Roman Perik-Zavodskii posted his RNA sequencing data in another database: Gene Expression Omnibus - escrow code GSE231654.

**Funding:** Sennikov SV grant number No. 21-15-00087 Russian Science Foundation https://rscf.ru/en/project/21-15-00087/ NO.

**Competing interests:** The authors have declared that no competing interests exist.

of circulating erythrocytes for respiratory gas exchange, but these cells involve other functions, including interaction with the immune system [4].

Many populations of induced or immature cells with immunoregulatory properties are considered as candidates for cellular immunotherapy in various pathological conditions: for example, regulatory T-cells [5], myeloid suppressor cells [6], mesenchymal stromal cells [7]. CD71+ erythroid cells, as immature cells, are usually absent or very rare in the blood of healthy adults, but are abundant in the spleen of newborn mice and in human umbilical cord blood and have immunoregulatory properties [8]. CD71+ erythroid cells provide feto-maternal tolerance; are responsible for the susceptibility of newborns to infections [9–11]; infiltrate the tumor and contribute to its progression [12]; accumulate in the peripheral blood during SARS-CoV-2 infection, leading to hypoxia, anemia, and coagulopathy in moderate to severe infection [13, 14]; reduce the production of pro-inflammatory cytokines by myeloid cells and reduce the activation and secretion of cytokines by lymphocytes and promote the production of anti-inflammatory cytokines and the differentiation of naive CD4+ T-cells into various types of T-cells, T-helper cells type 2 (Th2) [15], induced T-regulatory cells [8]. An analysis of the heterogeneity of erythroid precursors from the yolk sac, fetal liver, cord blood, and bone marrow in humans has shown that a population of immunoerythroid cells with dual erythroid and immunoregulatory networks is present at all stages of human ontogenesis. It is the immunomodulatory subgroup, and not the entire population of erythroid precursors, that may represent the cellular basis of immunoregulatory functions [16] However, the potential contribution of erythroid nucleated cells to non-respiratory physiological processes still requires detailed study. Therefore, studies aimed at further characterizing the immunological role of erythroid nucleated cells are of great importance. Since we assume that erythroid cells with pronounced immunoregulatory properties can be used for nonspecific immunotherapy, we conducted a study of the immunoregulatory properties of erythroblasts obtained in vitro from CD34-positive (CD34+) progenitor cells from the bone marrow.

## Materials and methods

### Donors

Bone marrow sampling from healthy adult donors was approved by the local ethics committee of the Research Institute of Fundamental and Clinical Immunology at meeting No. 129 on 17 February 2021. Each donor signed an approved informed consent form and underwent a general clinical examination (a general clinical blood test and electrocardiogram; a negative test for COVID-19, measurement of pressure and pulse immediately before the bone marrow sampling procedure). The age of the healthy bone marrow donors was $27.33 \pm 6.34$ years (mean ± standard error of the mean), and the study population consisted of three males and three females. All healthy donors for the study were recruited between February and December 2021. Bone marrow was obtained by puncture of the ilium. Mononuclear cells were isolated by the standard method via a Ficoll–Urografin gradient (Ficoll: PanEco, Moscow, Russia; Urografin: Schering AG, Germany). For this purpose, the bone marrow was diluted with Dulbecco's solution (Biolot, St. Petersburg, Russia) at a ratio of 1:1, layered on the Ficoll–Urografin mixture (PanEco, Moscow, Russia) at a Ficoll/bone marrow ratio of 1:3, and centrifuged for 25 min at 312 RCF (Elmi, Latvia). Cells of the interphase ring were collected and washed twice with Dulbecco's solution.

### Cell culture

CD34+ cells were isolated from the obtained mononuclear cells through positive magnetic selection by means of the CD34 MicroBead kit Ultra Pure Kit (MicroBeads conjugated to

mouse monoclonal anti-human CD34 antibodies, isotype mouse IgG1, Cat# 130-046-702, Miltenyi Biotec, Germany); the CD34+ cells were next used to prepare induced CD71+ erythroid cells as described before [17]. Namely, CD34+ cells (10 000/ml) were cultured for 7 days (expansion stage) in the X-VIVO 15 serum-free medium (Lonza, Switzerland) supplemented with 25 ng/ml rhSCF (Biolegend, San Diego, United States), 50 ng/ml rhTPO (Biolegend, San Diego, United States), 50 ng/ml Flt3 ligand (Gibco, Thermo Fisher Scientific, USA), and 10 μl/ml of the 100x sodium selenite-insulin-transferrin (ITS) growth supplement (Biolot, St. Petersburg, Russia).

Cell differentiation was implemented in the next stage of cell cultivation. To this end, the cells ($2 \times 10^5$/ml) were cultivated for 3 days with supplementation of 3 IU/ml erythropoietin (the State Research Institute of Highly Pure Biopreparations, affiliated with the Federal Medico-Biological Agency, St. Petersburg, Russia), 25 ng/mL rhSCF (Biolegend, San Diego, United States), 10 ng/mL rhIL-3 (Biolegend, San Diego, United States), and 10 ng/ml rhIL-6 (Biolegend, San Diego, United States). The third stage (cell maturation) involved culturing $1*10^4$ cells/ml in the new portion of the medium X-VIVO supplemented with erythropoietin. The maturation stage continued for the next 5 days. After that, we took the supernatant for cytokine production analysis, and the cells were used for gene expression analysis and phenotyping. Cells were washed twice after each stage of culturing. We placed the cells in a fresh portion of the culture medium at each subsequent stage of cultivation. We collected the cell supernatant to determine the content of cytokines after the third stage of cultivation. During this period, cells were cultured in a medium supplemented with erythropoietin, which is absent in the Bio-Plex Pro Human Cytokine 48-Plex Screening Panel (Bio-Rad Laboratories, USA), so there was no background effect of cytokines that we added to differentiate erythroid cells.

CD34-negative (CD34–) bone marrow mononuclear cells were used to isolate CD71+ erythroid nucleated cells by positive magnetic selection for marker CD71 (MicroBeads conjugated to mouse monoclonal anti-human CD71 antibodies, clone: AC108.1, isotype mouse IgG2a, Cat# 130-046-201, Miltenyi Biotec, Germany). Next, we took $2*10^5$ cells after magnetic sorting for RNA isolation. The other part of the CD71+ erythroid cellscells was at a concentration of $1*10^6$ cells/ml and cultured for 72 hours in X-VIVO 15 medium supplemented with 10 μl/ml of 100x insulin transferrin. The culture supernatant was collected for analysis of immunoregulatory factors. and for a mixed lymphocyte reaction (see below) as a conditioned medium.

## Flow cytometry

By this procedure, the resultant cells were analyzed for markers CD45 Brilliant Violet 510 (mouse monoclonal antibodies against human antigen CD45, clone HI30, Isotype Mouse IgG1, κ, Cat# 304036, Biolegend, San Diego, United States), CD71-PE (mouse monoclonal antibodies against human antigen CD71, clone CY1G4, Isotype Mouse IgG2a, κ, Cat# 334106, Biolegend, San Diego, United States), and CD235a-FITC (mouse monoclonal antibodies against human antigen CD235a (Glycophorin A), clone HI264, Isotype Mouse IgG2a, κ, Cat# 349104, Biolegend, San Diego, United States). Antibodies to markers CD45-PerCP/Cyanine5.5 (mouse monoclonal antibodies against human antigen CD45, clone HI30, Isotype Mouse IgG1, κ, Cat# 304028, Biolegend, San Diego, United States), CD3-FITC (mouse monoclonal antibodies against human antigen CD3, clone OKT3, Isotype Mouse IgG2a, κ, Cat# 317306, Biolegend, San Diego, United States), CD4-PE/Cyanine7 (mouse monoclonal antibodies against human antigen CD4, clone RPA-T4, Isotype Mouse IgG1, κ, Cat# 300512, Biolegend, San Diego, United States), CD8a-APC/Cyanine7 anti-human (mouse monoclonal antibodies against human antigen CD8, clone HIT8a, Isotype Mouse IgG1, κ, Cat# 300926, Biolegend, San Diego, United States), CD16-PE (mouse monoclonal antibodies against human antigen

CD16, clone B73.1, Isotype Mouse IgG1, κ, Cat# 360704, Biolegend, San Diego, United States), and CD19-APC (mouse monoclonal antibodies against human antigen CD19, clone HIB19, Isotype Mouse IgG1, κ, Cat# 302212, Biolegend, San Diego, United States) were used to detect lymphoid-cell markers. For staining by flow cytometry, all antibodies, except for CD 235a, were used at a dose of 5 µl per 1 million cells in 100 µl of cell suspension. For the CD 235a antibody, we used 0.5 µg per 1 million cells in 100 µl of cell suspension. These doses of antibodies were recommended by the manufacturer. The analysis was performed on an Attune NxT Flow Cytometer (Thermo Fisher Scientific, USA).

## Gene expression profiling by means of the nanostring nCounter GX human immunology v2 kit

Total RNA was isolated from a 200,000-cell sample with the Total RNA Purification Plus Kit (Norgen Biotek, Canada). Concentration and quality of RNA in each sample were assessed on a Nanodrop 2000 spectrophotometer (Thermo Fisher Scientific, USA). All the RNA samples were diluted to 10 ng/µl with nuclease-free water, and the diluted total RNA was frozen at –80˚C until analysis. Gene expression profiling with the help of the Nanostring nCounter SPRINT Profiler analytical system was performed on 50 ng of total RNA from each sample. The RNA samples were analyzed by means of the nCounter Human Immunology v2 panel, which consists of 579 immune and inflammation-associated genes, 15 housekeeping genes, and eight negative and six positive controls; these controls were utilized to subtract background signals and to normalize the numbers of unprocessed mRNA transcripts in all samples. The samples were subjected to an overnight hybridization reaction at 65˚C, where 5–12 µl of total RNA was combined with 3 µl of nCounter Reporter probes, 0–7 µl of DEPC-treated water, 10 µl of hybridization buffer and with 5 µl of nCounter capture probes (total reaction volume was 30 µl). After the hybridization of the probes to targets of interest in the samples, the number of target molecules was determined on the nCounter digital analyzer and evaluated on the nSolver platform. The gene expression profiles were examined in the nSolver 4. nSolver 4 was used to perform QC and to perform normalization using added synthetic positive controls and the 15 housekeeping genes included in the panel. Then background thresholding was performed on the normalized data. The background thresholding procedure returned 50 genes with above noise level expression. The noise level was determined as: mean of negative controls + 2 standard deviations + 40

We performed differential gene expression using multiple T-tests (with Q < 0.05) in GraphPad Prism 9.4. The Volcano- and MA-plots were created in GraphPad Prism 9.4. The GSEA was done using GSEApy (https://github.com/zqfang/GSEApy). The heatmap was created using bioinfokit (https://github.com/reneshbedre/bioinfokit).

## Cytokine secretion analysis

Assessment of cytokine secretion (concentrations) in culture supernatants of CD71+ erythroid cells was performed using the Bio-Plex Pro Human Cytokine 48-Plex Screening Panel (Bio-Rad Laboratories, USA). This assay platform is based on fluorescent-bead technology combined with a sandwich immunoassay, allowing for stand-alone and multiplex assays of up to 100 analytes in a single microtiter plate well. For this assay, a panel of 47 cytokines was employed in 96-well microtiter plate format. Contents of each well were analyzed using the Bio-Plex 100 System (Bio-Rad Laboratories) to quantify each specific reaction on the basis of the color of the particles and fluorescent-signal intensity. The data were finally processed in the Bio-Plex Manager software (version 6.1) by five-parameter curve fitting and were converted to picograms per milliliter.

## Mixed lymphocyte reaction

Peripheral blood was drawn from donors into vacuum tubes containing EDTA as an anticoagulant (Improvacuter, China). Peripheral blood mononuclear cells (PBMCs) were isolated from whole blood samples using a conventional Ficoll–Urografin density gradient method. Briefly, peripheral blood was diluted with an equal volume of RPMI-1640 medium (Biolot, St. Petersburg, Russia), then layered on a Ficoll-Urografin solution ($\rho$ = 1.077 g/L) and centrifuged at 400× g and room temperature for 40 min. Mononuclear cells were collected from an opalescent layer located at the phase boundary over the entire tube cross-section. The RPMI-1640 medium supplemented with 10% fetal calf serum (FCS) (Biowest, Nuaillé, France), 2 mM L-glutamine (Biolot, Saint Petersburg,Russia), $5 \times 10{-}4$ M 2-mercaptoethanol (Sigma-Aldrich, St. Louis, MO, USA), 25 mM HEPES (Biolot, Saint Petersburg, Russia), 80 μg/mL gentamicin (KRKA, Novomesto, Slovenia) and 100 μg/mL benzylpenicillin (Biolot, Saint Petersburg, Russia) (hereinafter designated the culture medium) was used for PBMCs cultivation. "Effector" cells were dispensed into wells at $10^5$ cells/well. To block proliferative activity of "inducer" cells, they were treated with mitomycin C (Sigma, Germany) at 50 μg/ml for 60 min, washed three times with the culture medium, and were added in the amount of $10^5$ cells/well to the "effectors." The cells were incubated in 200 μl of the culture medium at 37˚C, 5% CO2, and 100% humidity for 4 days. This assay was carried out in 96-well round-bottom microtiter plates (TPP, Switzerland), in three identical wells with similar samples analyzed in parallel. The conditioned medium (culture supernatant) from the studied CD71+ erythroid cells constituted half of the total volume of a well during the cultivation. The proliferation rate of the cultured cells was assessed by means of the WST reagent (Takara Bio, Japan), which was added at 4 h before the end of cultivation. The results are presented as optical density.

## Statistic

Statistical analysis of the data to find differences between groups was conducted in the GraphPad Prism 8.4 and 9.4 software. Cytokine production was evaluated with the CytokineExplore online tool (http://exabx.com/apps/cytokineexplore/). The Mann–Whitney test was performed to compare the production of cytokines between the different types of CD71+ erythroid cells. ANOVA with multiple comparison criteria was carried out to evaluate statistical significance of the differences between the cell groups. Results were assumed to be statistically significant at P-value < 0.05, and the data are presented as a median with an interquartile range.

## Results

### Phenotypic characteristics of the induced CD71+ erythroid cells

CD71+ erythroid cells are a heterogeneous population with several stages of differentiation. The cells of each stage realize their unique functions, which are difficult to separate at the level of the whole organism. We determined the phenotype and maturation stage of CD71+ erythroid cells using the expression of CD 71, CD 235a and FSC [18, 19]. According to these sources, we differentiated proerythroblasts as CD235a$^{low}$CD71$^{high}$ cells, basophilic erythroblasts as CD 235$^{high}$ CD71$^{pos}$ FSC$^{high}$ cells, polychromatophilic erythroblasts as CD235a$^{high}$CD 71$^{pos}$FSC$^{middle}$, orthochromatophilic erythroblasts/reticulocytes as CD235$^{high}$CD71$^{pos}$FSC$^{low}$, erythrocytes CD235a$^{high}$CD71$^{neg}$. The population of orthochromatophilic erythroblasts/reticulocytes CD 235$^{high}$CD71$^{pos}$ FSC$^{low}$ was clearly divided into two parts by FSC, so we were able to differentiate orthochromatophilic erythroblasts and reticulocytes from each other. It was found that induced erythroblasts can be divided into a CD45+ population and a CD45–

**Table 1. Proportions (%) of CD45+ and CD45- cells among the induced CD71+ erythroid cells derived from CD34+ bone marrow cells and bone marrow CD71+ erythroid cells.** The data are presented as the mean ± standard derivation (n = 6).

| | Bone marrow erythroblasts | Induced erythroblasts |
|---|---|---|
| CD45+ cells | 2.47±0.6 | 46.74±9.28 |
| CD45− cells | 97.53±0.6 | 52.59±9.31 |

population (Table 1). The distribution of natural bone marrow CD71+ erythroid cells and induced CD71+ erythroid cells is shown in Figs 1 and 2, respectively.

We compared the content of CD71+ erythroid cells of different stages of development for natural and induced CD71+ erythroid cells (Fig 3).

Our data show that in the bone marrow of a healthy adult, polychromatophilic erythroblasts predominate. The general population of induced erythroblasts differs from bone marrow cells by a decrease in the content of polychromatophilic erythroblasts and the redistribution of all other stages towards the accumulation of more mature forms. At the same time, basophilic and polychromatophilic erythroblasts predominate among CD 45-positive CD71+ erythroid cells, while orthochromatophilic and reticulocytes predominate among CD 45-negative ones. Thus, induced CD71+ erythroid cells retain their phenotypic identity with bone marrow CD71+ erythroid cells, but their distribution by stages differs markedly.

To determine the feasibility of deriving a lymphoid lineage from the CD34+ cells, we quantified the expression of markers CD3, CD4, CD8, CD16, and CD19. Induced CD71+ erythroid cells from bone marrow CD 34+ cells had the phenotype CD45+CD3–CD4–CD8–CD16–CD19– and CD45–CD3–CD4–CD8–CD16–CD19– (Fig 4).

## Gene expression analysis

We examined the expression of 579 genes associated with inflammation and immune response. Analysis of the expression profile of natural and induced erythroblasts revealed differential expression of 49 genes (Fig 5).

Most differentially expressed genes are related to pathogen interaction processes, innate immune system, cytokine signaling, adaptive immune system, lymphocyte activation, hemostasis, apoptosis, T- and B-cell signaling. The minimum numbers of differentially expressed genes are related to autophagy processes, Th17-, Th1-, Treg-differentiation, signaling through TGF-b and Type I Interferon.

As studied cells had obvious transcriptomic heterogenecity we performed differential gene expression analysis. To do this, we built Volcano- and MA- plots of differentially expressed genes A volcano plot (Fig 6) is a type of scatterplot that shows statistical significance (P-value) versus magnitude of change (fold change). It enables quick visual identification of genes with large fold changes that are also statistically significant. These may be the most biologically significant genes. In a volcano plot, the most upregulated genes are towards the right, the most downregulated genes are towards the left, and the most statistically significant genes are towards the top. MA-plot (Fig 7) is a 2-dimensional (2D) scatter plot used for visualizing gene expression datasets MA plot visualize and identify gene expression changes from two different conditions in terms of log fold change on Y-axis and log of the mean of normalized expression counts of two conditions on X-axis. Generally, genes with lower mean expression values will have highly variable log fold changes. Genes with similar expression values will cluster around 0 value i.e genes expressed with no significant differences in between sampes. Points away from 0 line indicate genes with significant expression, For example, a gene is upregulated and downregulated if the point is above and below 0 line respectively.

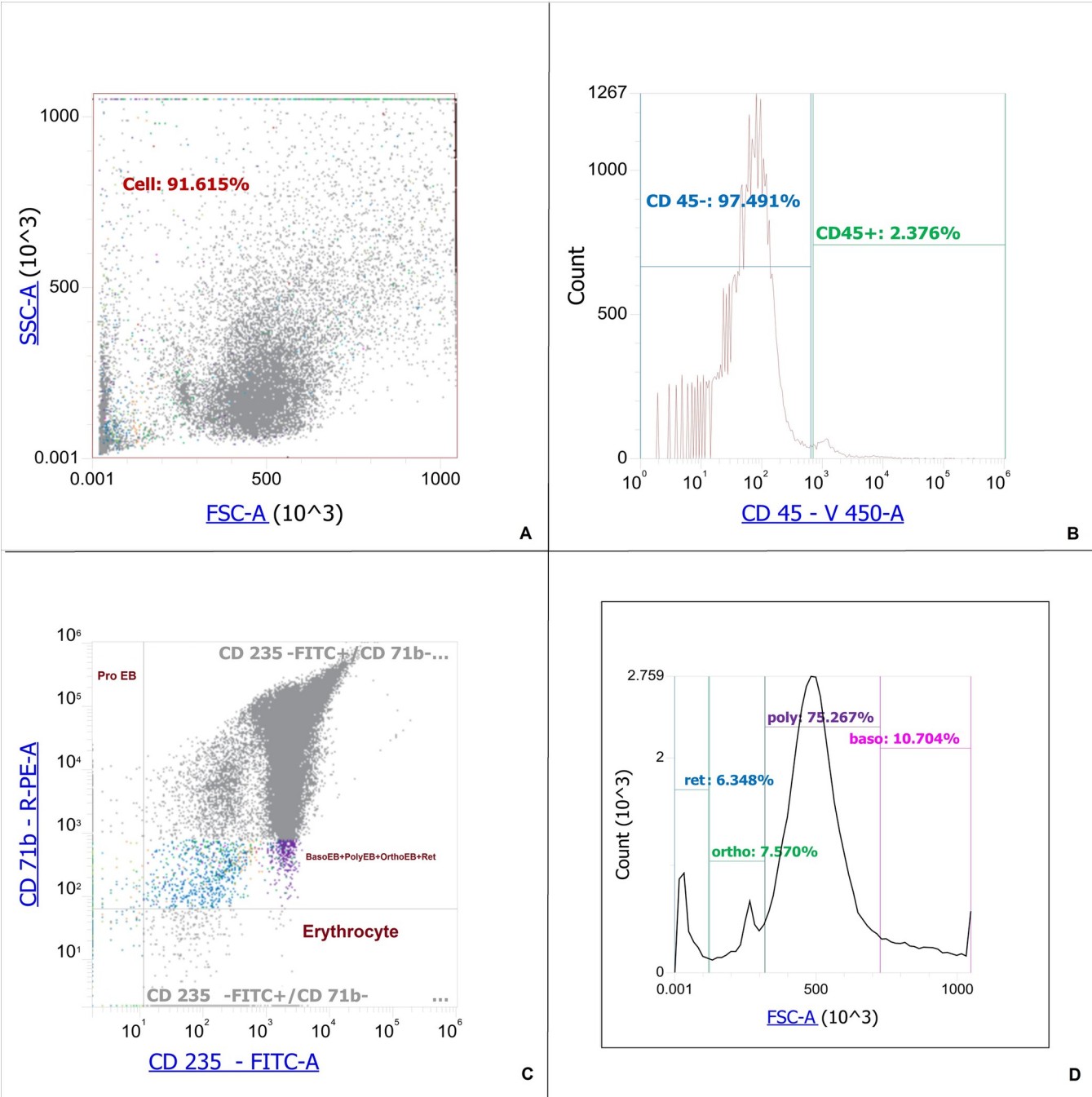

**Fig 1. Staging of natural bone marrow CD71+ erythroid cells.** A–Dot-plot for natural bone marrow CD71+ erythroid cells. B–Histogram of the distribution of bone marrow CD71+ erythroid cells by marker CD 45. C–Dot-plot of the distribution of bone marrow CD71+ erythroid cells by marker CD 71 and CD 235a from CD 45-negative natural bone marrow CD71+ erythroid cells, ProEB–proerythroblasts (CD235a$^{low}$CD71$^{high}$ cells), BasoEB+PolyEB+OrthoEB+Ret– basophilic erythroblasts, polychromatophilic erythroblasts, orthochromatophilic erythroblasts/reticulocytes (CD235$^{high}$CD 71$^{pos}$ cells), Erythrocytes– CD235a$^{high}$CD71$^{neg}$. D–Histogram of the distribution of stages of development of erythroid cells with a phenotype CD235$^{high}$CD 71$^{pos}$ depending on the FSC. Ret–reticulocytes (CD 235a$^{high}$CD71$^{pos}$ FSC$^{low}$), Ortho–orthochromatophilic erythroblasts (CD235a$^{high}$CD71$^{pos}$FSC$^{low}$ cells), Poly–polychromatophilic erythroblasts (CD235a$^{high}$CD 71$^{pos}$FSC$^{middle}$ cells), Baso–basophilic erythroblasts (CD 235a$^{high}$ CD71$^{pos}$ FSC$^{high}$ cells).

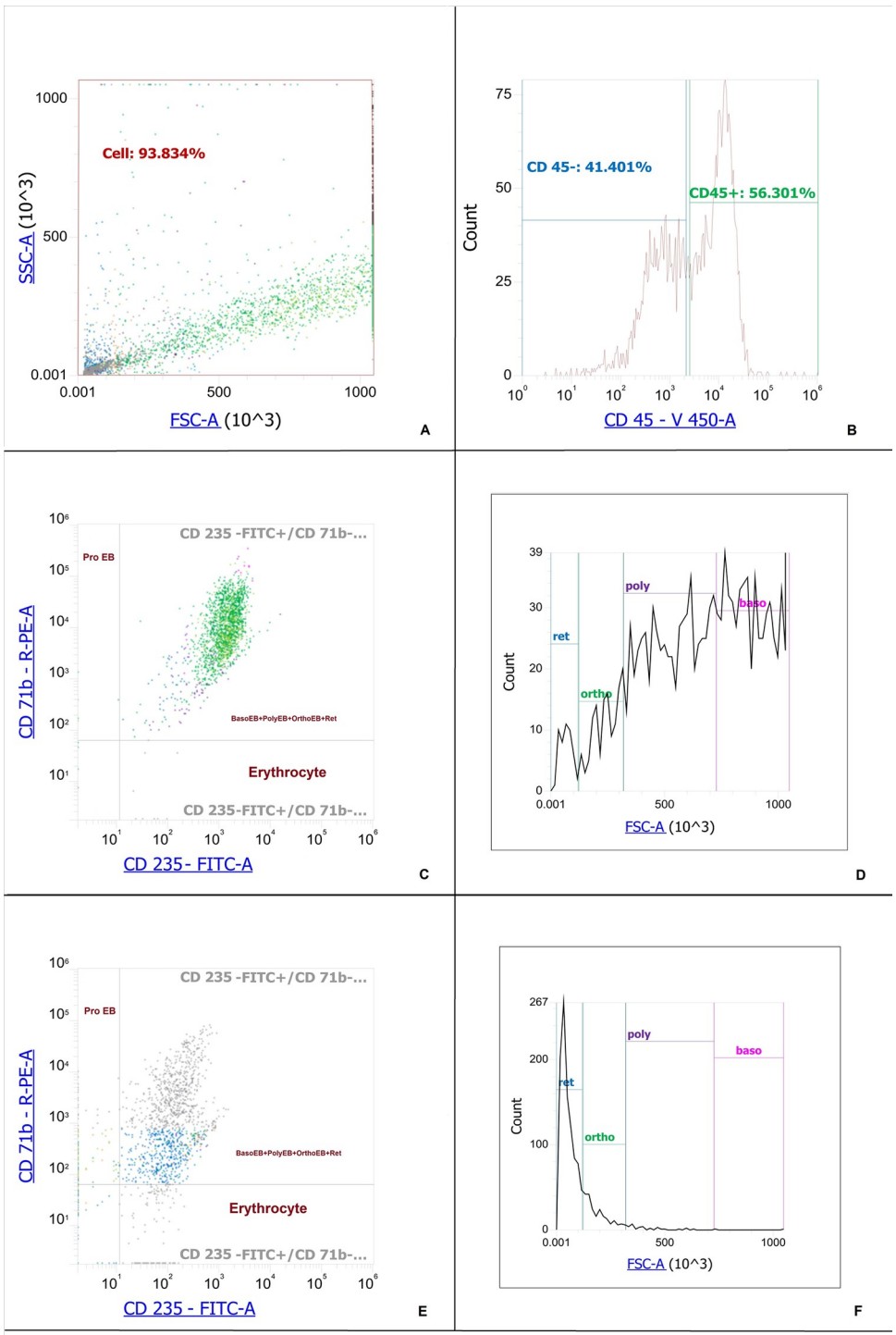

**Fig 2. Staging of induced CD71+ erythroid cells derived from CD34+ bone marrow cells.** A–Dot-plot for induced CD71+ erythroid cells derived from CD34+ bone marrow cells. B–Histogram of the distribution of induced CD71 + erythroid cells derived from CD34+ bone marrow cells by marker CD 45. C–Dot-plot of the distribution of bone marrow CD71+ erythroid cells by marker CD 71 and CD 235a from CD 45-positive induced CD71+ erythroid cells. D–Histogram of the distribution of stages of development of CD71+ erythroid cells with a phenotype CD45$^{pos}$CD235a$^{high}$CD 71$^{pos}$ depending on the FSC. E–Dot-plot of the distribution of bone marrow CD71+ erythroid by marker CD 71 and CD 235a from CD 45-negative induced CD71+ erythroid cells. F–Histogram of the distribution of stages of development of CD71+ erythroid cells with a phenotype CD45$^{neg}$CD235a$^{high}$CD 71$^{pos}$ depending on the FSC, ProEB–proerythroblasts (CD235a$^{low}$CD71$^{high}$ cells), BasoEB+PolyEB+OrthoEB+Ret–basophilic erythroblasts,

polychromatophilic erythroblasts, orthochromatophilic erythroblasts/reticulocytes (CD235a$^{high}$CD 71$^{pos}$ cells), Erythrocytes–CD235a$^{high}$CD71$^{neg}$, Ret–reticulocytes (CD 235a$^{high}$CD71$^{pos}$ FSC$^{low}$), Ortho–orthochromatophilic erythroblasts (CD235a$^{high}$CD71$^{pos}$FSC$^{low}$ cells), Poly–polychromatophilic erythroblasts (CD235a$^{high}$CD 71$^{pos}$FSC$^{middle}$ cells), Baso–basophilic erythroblasts (CD 235$^{high}$ CD71$^{pos}$ FSC$^{high}$ cells).

The induced CD71+ erythroid cells are characterized by the predominance of the expression of B2M, CD44, HLA-A, IL-8 genes, which are involved in the processes of lymphocyte activation, cell adhesion, chemokine signaling, NLR signaling, type II Interferon Signaling, type I Interferon Signaling. CD71+ erythroid cells of the bone marrow are characterized by the predominance of expression of the ILF3, TAL1, and CD36 genes, which are involved in the regulation of transcription and oxidative stress.

Analysis of the MA plot showed that CD71+ induced erythroid cells grown in vitro show significantly lower expression of early erythroblast-associated markers CD36, ITGA4 and TFRC and significantly higher expression of the IL-8 chemokine.To characterize the potential functions altered by DEG, we performed an enrichment analysis of GO pathways.using the

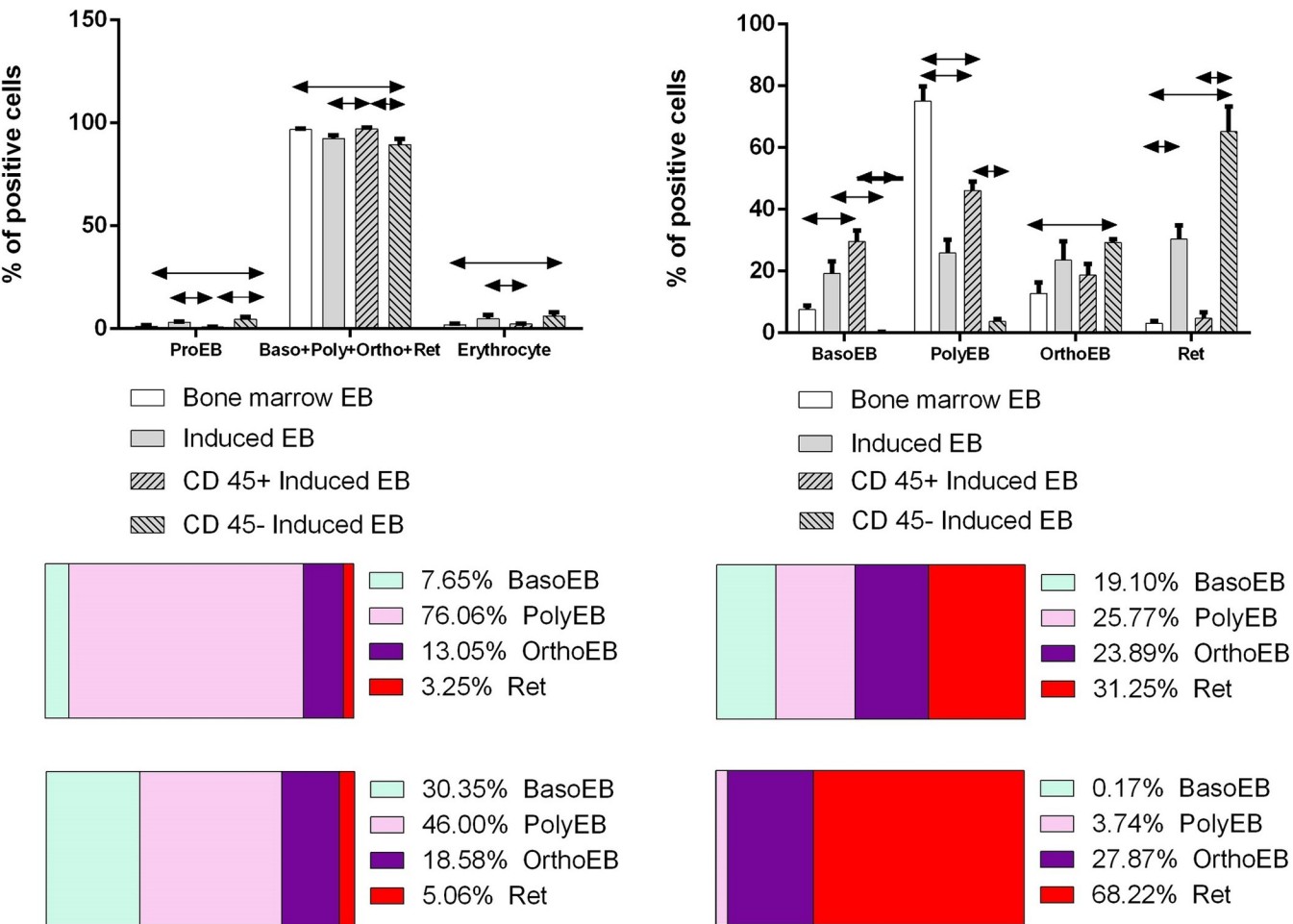

**Fig 3. Comparison of the distribution of stages of development of bone marrow CD71+ erythroid cells and CD71+ erythroid cells induced from CD34 + bone marrow cells.** Kruskal-Wallis test ANOVA with Dunn's correction for multiple comparisons was carried out to evaluate statistical significance of the differences between the cell groups. Results were assumed to be statistically significant at p-value < 0.05, and the data are presented as a median with an interquartile range.

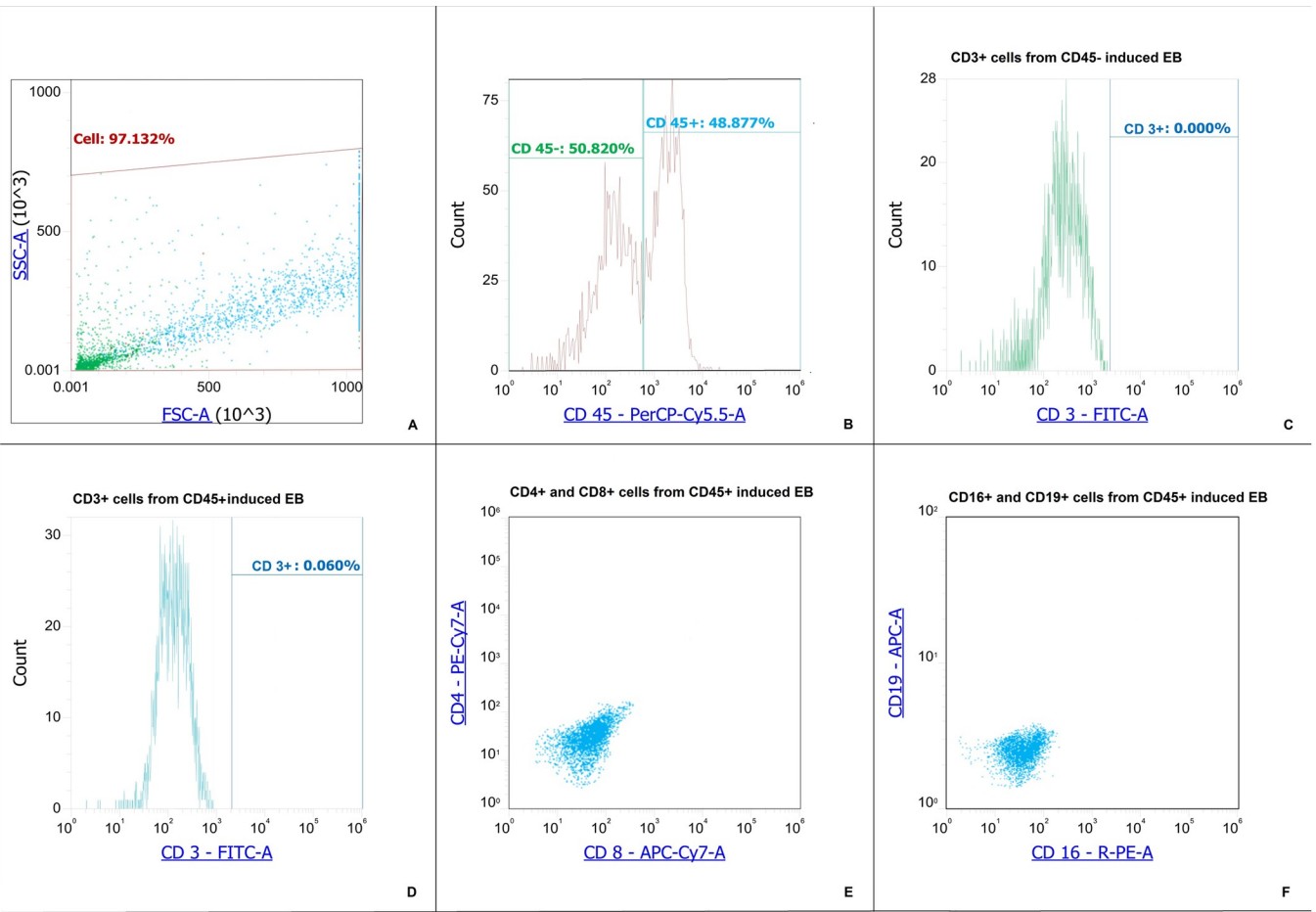

**Fig 4. Expression of lymphoid cell markers on the surface of induced CD71+ erythroid cells.** A–Dot-plot for induced CD71+ erythroid cells derived from CD34+ bone marrow cells. B–Histogram of the distribution of induced CD71+ erythroid cells derived from CD34+ bone marrow cells by marker CD 45. C – Histogram of the distribution of induced CD71+ erythroid cells derived from CD34+ bone marrow cells by marker CD 3 from CD 45-negative induced CD71 + erythroid cells. D–Histogram of the distribution of induced CD71+ erythroid cells derived from CD34+ bone marrow cells by marker CD 3 from CD 45-positive induced CD71+ erythroid cells. E–Dot-plot of the distribution of bone marrow CD71+ erythroid cells by marker CD 8 and CD 4 from CD 45-positive induced erythroblasts. F–Dot-plot of the distribution of bone marrow CD71+ erythroid cells by marker CD 16 and CD 19 from CD 45-positive induced erythroblasts.

"Gene Ontology Biological Process" terms (Fig 8). Notable terms were the "regulation of hematopoietic stem cell differentiation" and the "regulation of hematopoietic progenitor cell differentiation".

## Analysis of cytokine production by natural and induced CD71+ erythroid cells

We investigated the production of 47 cytokines in conditioned media of natural bone marrow CD71+ erythroid cells and induced CD71+ erythroid cells. Partial least squares discriminant analysis (PLS-DA) and an importance plot of variables for two groups [20] were employed to identify differences in the production of cytokines between the two groups of CD71+ erythroid cells under study. Partial Least Squares Discriminant Analysis (PLSDA) makes a graph showing the projection multi-cytokine data on only two axes/components. The two groups are colored and symbolized differently. PLSDA is used when using partial least squares regression to

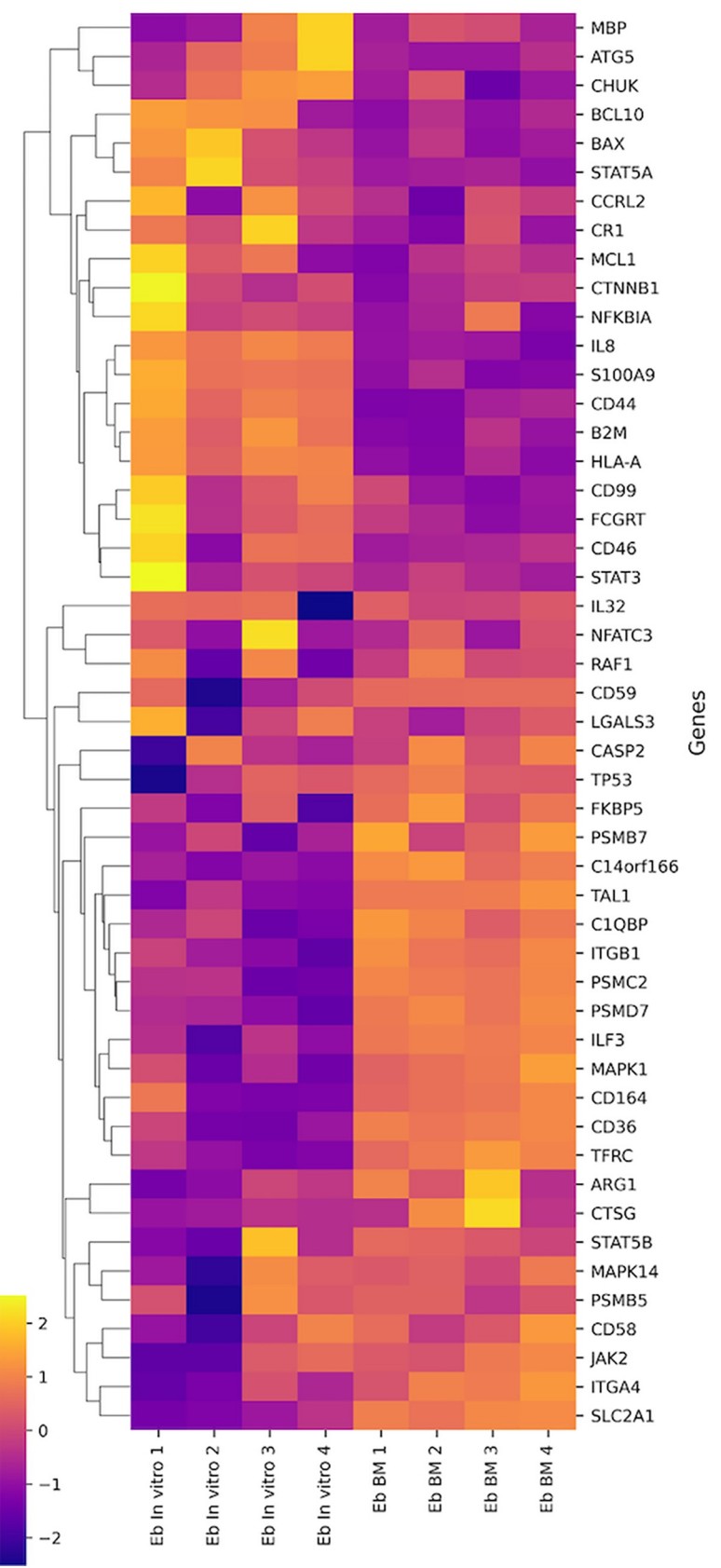

**Fig 5. Heat map of studied cells' transcriptome.**

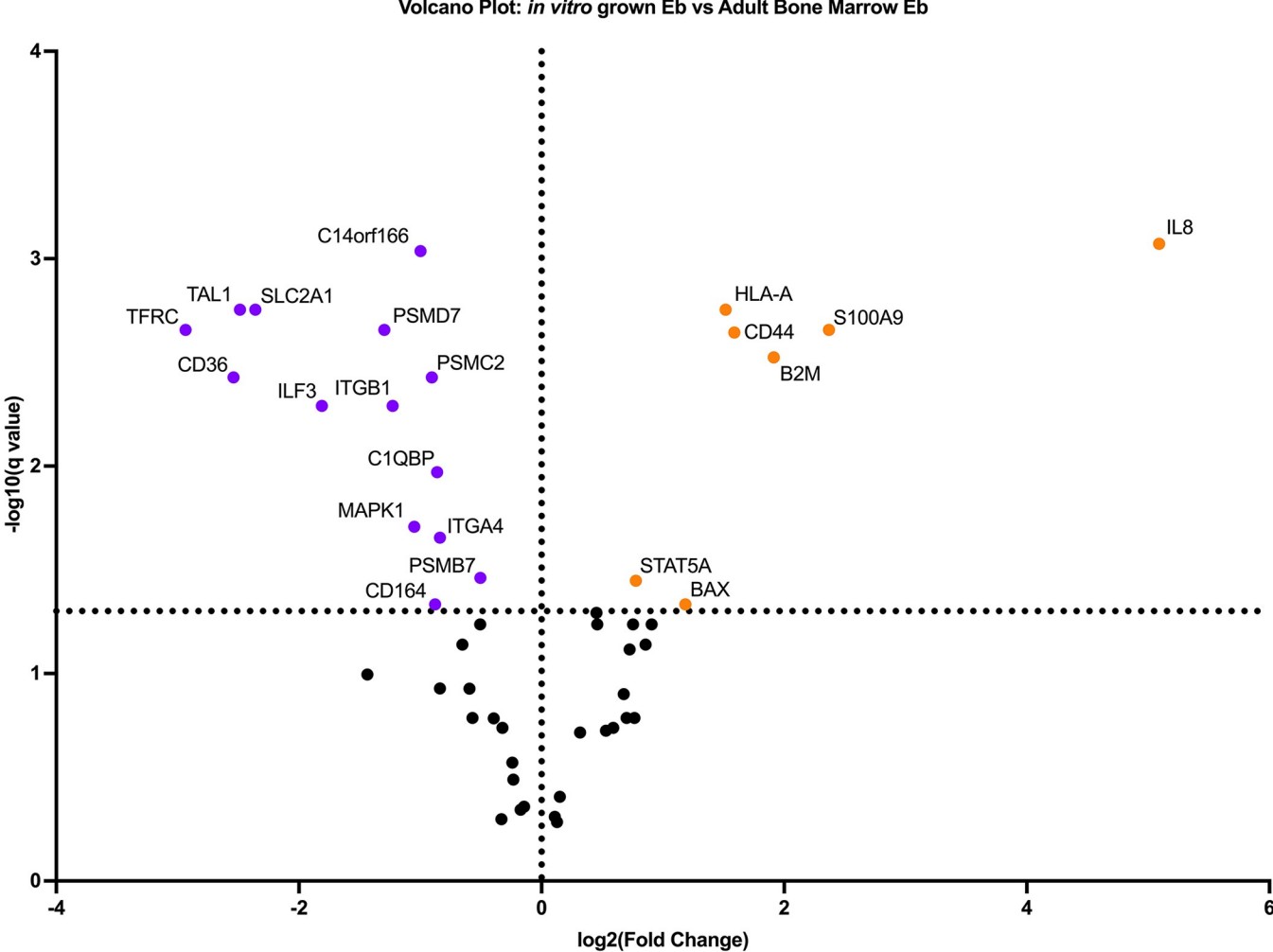

**Fig 6. Volcano-plots of differentially expressed genes.** The purple dots correspond to the samples of induced CD71+ erythroid cells, the orange dots correspond to the samples of natural CD71+ erythroid cells of bone marrow.

problems with categorical outcomes (in our case, induced versus natural erythroblasts). Axles are marked LV1 (latent variable 1) and LV2 (latent variable 2). If the data points of the two groups overlap strongly, then this implies that the two groups have a similar cytokine profile. On the other hand, if the data points barely overlap, then this would mean that groups have very different profiles. As a precaution, if the number of samples is very small, and the number cytokines is high, the PLSDA plot may exaggerate the difference between groups. To this end, we subdivided all cytokines into functional groups: chemokines (10 cytokines: CTACK, Eotaxin, GRO-alpha, IP-10, MCP-1, MCP-3, MIF, MIG, MIP-1alpha, RANTES), interleukins (21 cytokines: IL-1alpha, IL-1betta, IL-1RA, IL-2, IL-2Ralpha, IL-3, IL-4, IL-5, IL-6, IL-7, IL-8, IL-9, IL-10, IL-12 (p70), IL-12 (p40), IL-13, IL-15, IL-16, IL-17, IL-18, LIF), growth factors (8 cytokines: basic FGF, HGF, betta-NGF, PDGF-BB, SCF, SCGF-betta, SDF-1alpha, VEGF), colony-stimulating factors (3 cytokines: G-CSF, GM-CSF, M-CSF), interferons (2 cytokines: IFN-alpha2, IFN-gamma) and TNF family cytokines (3 cytokines: TNF-alpha, TNF-betta, TRAIL) to reduce the number of false positive results in the context of the small number of samples and the large number of analytes. Data points of the production of chemokines, interleukins,

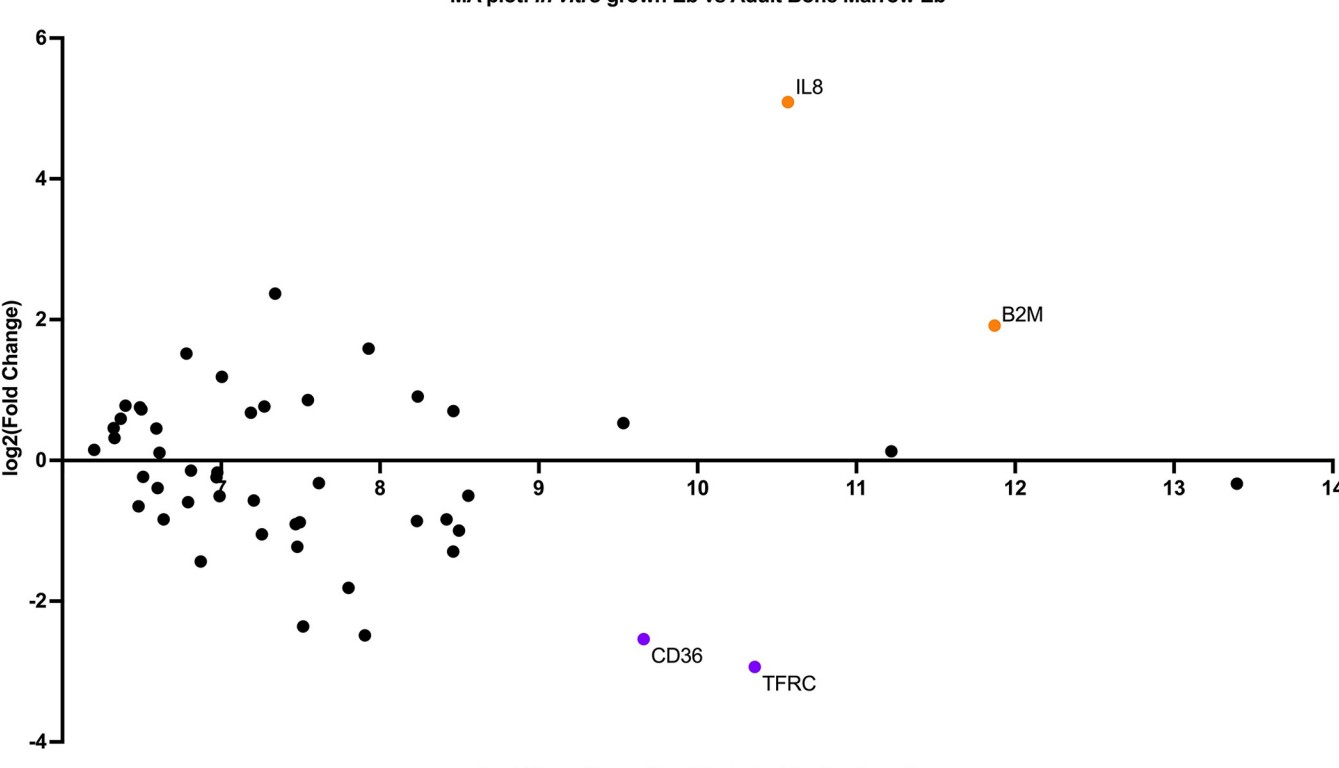

**Fig 7. MA-plots of differentially expressed genes.** The purple dots correspond to the samples of induced CD71+ erythroid cells, the orange dots correspond to the samples of natural CD71+ erythroid cells of bone marrow.

and growth factors did not overlap between the two populations of CD71+ erythroid cells (natural and induced), indicating that the two cell groups have different cytokine production profiles (Fig 9). Data points of the production of colony-stimulating factors, interferons, and TNF family cytokines overlapped between the populations of the natural and induced CD71+ erythroid cells, suggesting that these cell groups have similar cytokine production profiles (Fig 10).

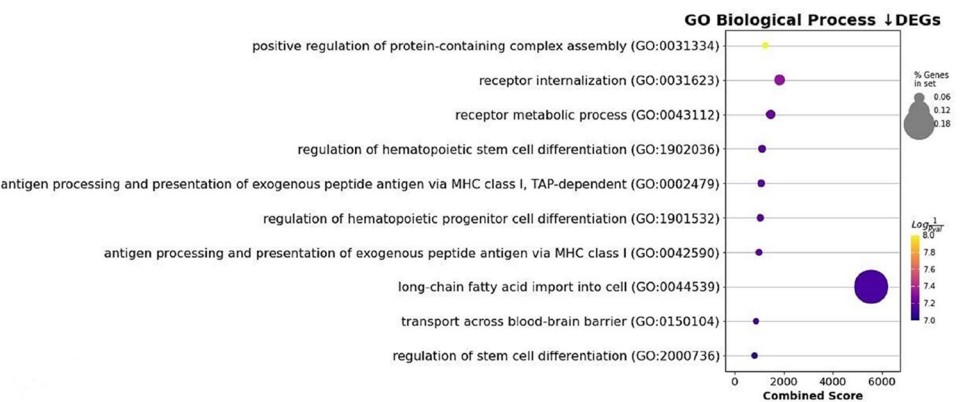

**Fig 8. Plot of Gene Set Enrichment Analysis (GSEA) results showing down-regulated differentially expressed genes.**

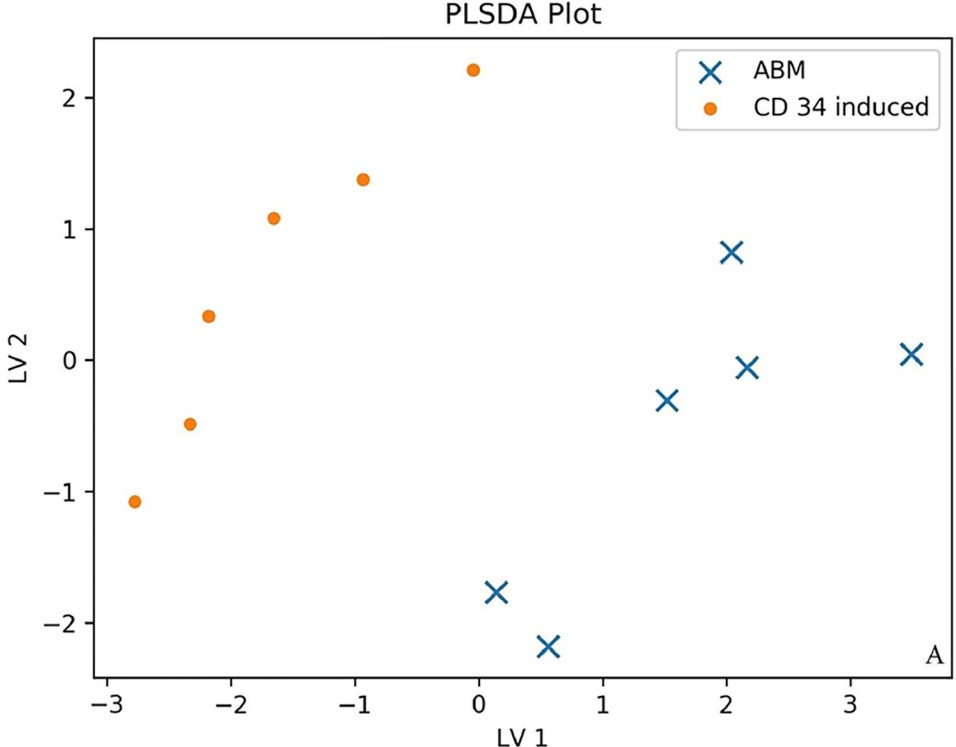

**Fig 9. The PLS-DA plot of cytokine production by different types of CD71+ erythroid cells (n = 6 for bone marrow CD71+ erythroid cells and induced CD71+ erythroid cells from CD34+ bone marrow cells).** An example distribution of data points for chemokines, interleukins, and growth factors is shown. (x)–ABM–bone marrow erythroblasts. (·)–CD 34 induced–induced erythroblasts from CD34+ bone marrow cells.

For 13 cytokines, we found significant differences in the production of cells by cells of both studied groups. In a comparison of the cytokine production between the two types of erythroblasts, it was demonstrated that the natural CD71+ erythroid cells from bone marrow secrete significantly more IL-1β, IL-13, RANTES, IL-12 (p40), IL-1α, IL-10, IL-17, IP-10, PDGF-BB, and SDF-1α, whereas the induced CD71+ erythroid cells secrete larger amounts of IL-3, SCF, and IL-5 (Figs 11 and 12).

## Effect of the environment of CD71+ erythroid cells on the mixed culture of lymphocytes

To assess functional properties of the mediators secreted by the two types of CD71+ erythroid cells, the mixed lymphocyte reaction was carried out next. The principle behind this assay is as follows: T-cells from one donor will proliferate in the presence of antigen-presenting cells from another donor. The reason is the recognition of an HLA mismatch between two unrelated donors, which triggers an immune response of T cells. Mixed cultivation of lymphocytes is often used as a way to induce nonspecific stimulation/activation of T cells in culture [21]. To test the effects on antigen-presenting and allostimulatory functions of T cells, we used conditioned media from our natural erythroblasts and induced erythroblasts.

In the experiment on the influence of the conditioned medium from the natural CD71+ erythroid cells, it was demonstrated that proliferative activity stayed at the same level as when the cells were stimulated with the alloantigen, i.e. processes of presentation, recognition, and response to the alloantigen remained intact (Fig 13). On the contrary, when the conditioned

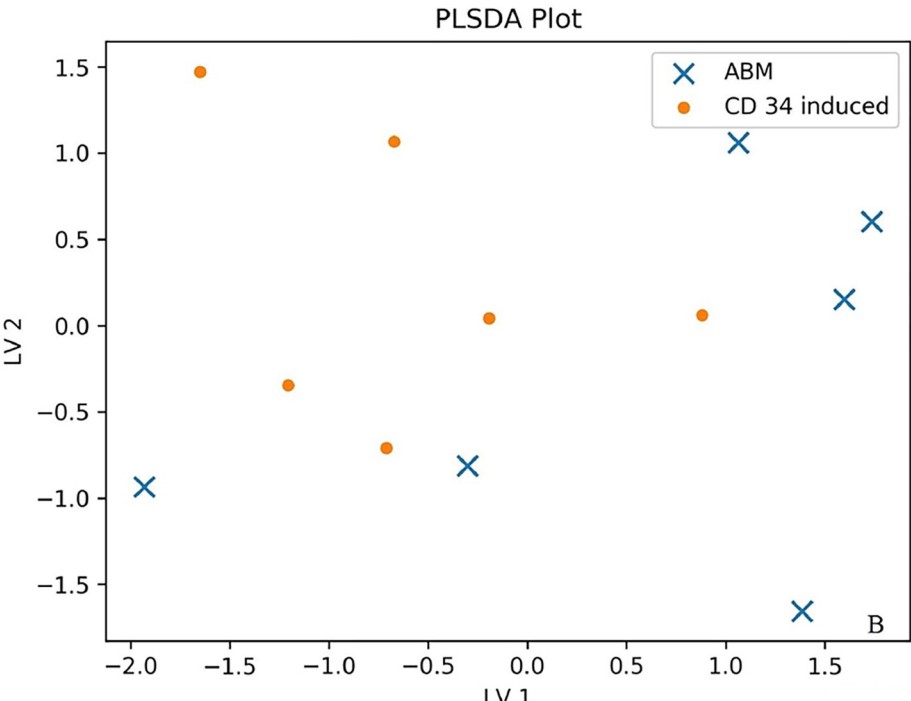

**Fig 10. The PLS-DA plot of cytokine production by different types of CD71+ erythroid cells (n = 6 for bone marrow CD71+ erythroid cells and induced CD71+ erythroid cells from CD34+ bone marrow cells).** An example distribution of data points for colony-stimulating factors, interferons, and TNF family cytokines is shown. (x)–ABM–bone marrow erythroblasts. (·)–CD 34 induced–induced erythroblasts from CD34+ bone marrow cells.

medium from the induced CD71+ erythroid cells was applied, the proliferative activity proved to be suppressed in the mixed lymphocyte culture (Fig 13).

## Discussion

In vitro cultivation of CD34+ bone marrow cells during incubation with erythropoietin (EPO), SCF, and IL-3 [22] makes it possible to obtain erythroblasts for further research or for the creation of possible immunotherapeutic technologies. Induced CD71+ erythroid cells derived from in vitro CD34+ precursors consist of two populations: CD45+CD71+CD235aa + cells and CD45−CD71+CD235a+ cells. CD 45+ CD71+ erythroid cells do not express lymphoid cell markers and are present throughout the entire erythroblast culture. Such CD45 + CD71+ erythroid cells are found in foci of extramedullary erythropoiesis in the spleen, where they suppress anti-infective and anti-tumor immunity in neonates and adults with malignancy. CD45+CD71+ erythroid cells suppress T cells due to the secretion of reactive oxygen species, IL-10 and TGF-β via the paracrine pathway and during intercellular contact, and their suppressor effect is more pronounced than that of suppressor cells of myeloid origin [23]. CD71+ erythroid cells expressing the CD45 panleukocyte marker at the earliest stages of differentiation are more powerful immunoregulators than CD45+ CD71+ erythroid cells at later stages [23, 24]. Currently, the presence of immunosuppressive properties of CD71+ erythroid cells with CD 45 expression is confirmed by their extramedullary presence in the bloodstream of patients with SARS-CoV-2 [25]. CD45+CD71+ erythroid cells downregulate effector CD8+ T cell activity (eg, granzyme B expression and degranulation [CD107] capacity) via

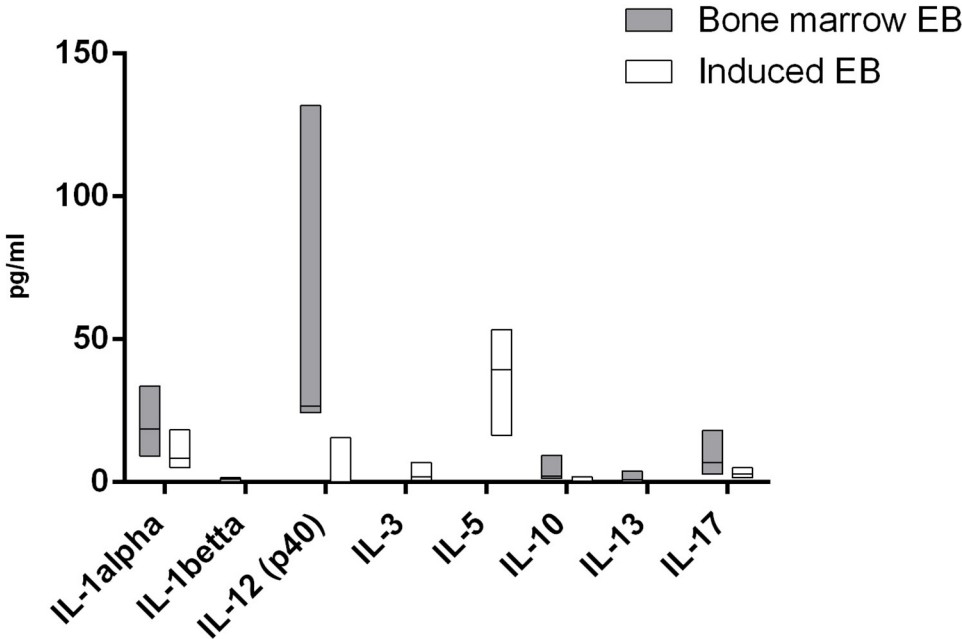

**Fig 11. Cytokines that manifested significant differences in secretion of interleukins between the two types of CD71+ erythroid cells (n = 6 for bone marrow CD71+ erythroid cells and induced CD71+ erythroid cells from CD34$^+$ bone marrow cells).** The Mann–Whitney test was performed to compare the production of cytokines between the different types of CD71+ erythroid cells. Results were assumed to be statistically significant at $p$-value $< 0.05$, and the data are presented as a median with an interquartile range.

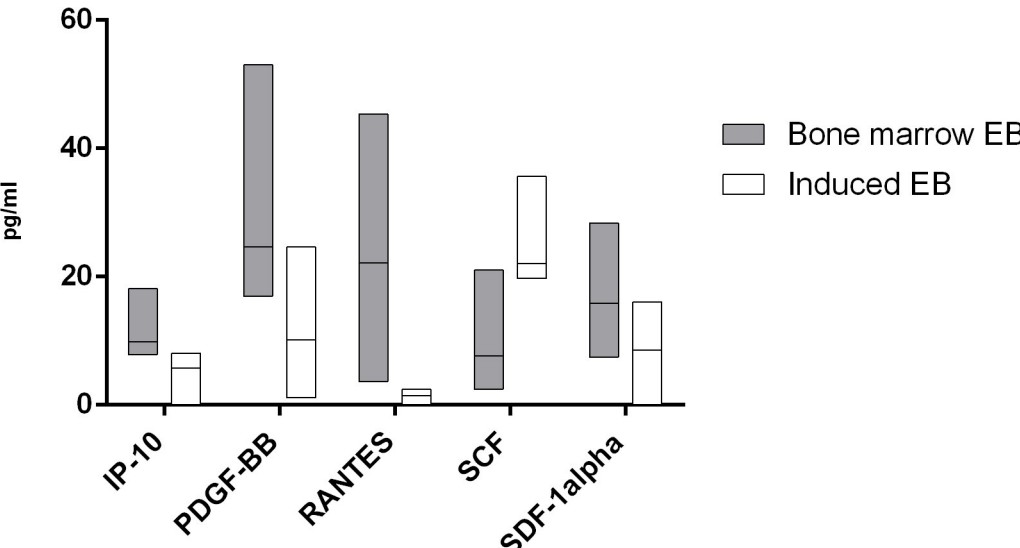

**Fig 12. Cytokines that manifested significant differences in secretion of chemokines and growth factors between the two types of CD71+ erythroid cells (n = 6 for bone marrow CD71+ erythroid cells and induced CD71+ erythroid cells from CD34$^+$ bone marrow cells).** The Mann–Whitney test was performed to compare the production of cytokines between the different types of CD71+ erythroid cells. Results were assumed to be statistically significant at $p$-value $< 0.05$, and the data are presented as a median with an interquartile range.

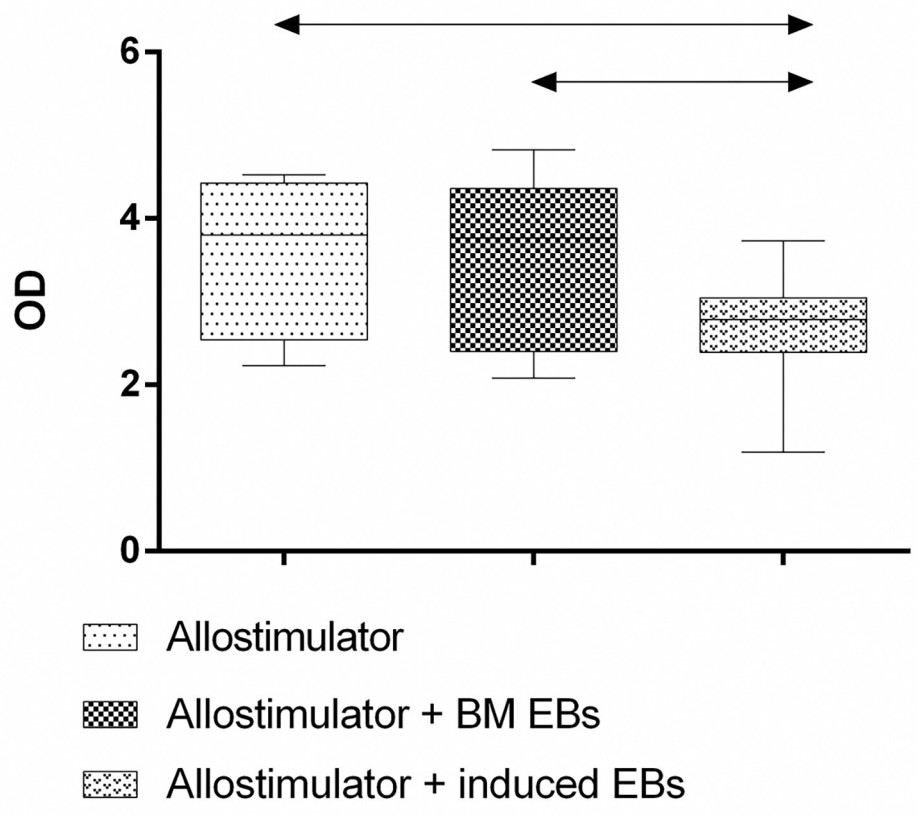

**Fig 13. Effects of the conditioned media from either the natural or induced bone marrow CD71+ erythroid cells on the mixed lymphocyte reaction (n = 4).** The results are presented as optical density of allogeneic cultures. Statistical analysis of the data was conducted in the GraphPad Prism 8.4 software. Two-way ANOVA with Tukey's correction for multiple comparisons was carried out to evaluate statistical significance of the differences between the cell groups. Results were assumed to be statistically significant at $p$-value < 0.05, and the data are presented as a median with an interquartile range. "Allostimulator": culture of mononuclear cells with the addition of allogeneic donor cells treated with mitomycin C; "Allostimulator + BM EBs": culture of mononuclear cells with the addition of mitomycin C–treated allogeneic donor cells and supplementation with the conditioned medium from natural bone marrow CD71+ erythroid cells; "Allostimulator + induced EBs": culture of mononuclear cells with the addition of mitomycin C–treated allogeneic donor cells and supplementation with the conditioned medium from the induced CD71+ erythroid cells derived from CD34+ bone marrow cells of healthy donors.

reactive oxygen species [26]. Our study confirmed that the conditioned medium of induced CD71+ erythroid cells has a more pronounced inhibitory effect on the allogeneic mixed lymphocytic reaction compared to the conditioned medium from natural bone marrow CD71 + erythroid cells. This conclusion can be explained by the report that human bone marrow contains much fewer early-stage CD71+ erythroid cells than late-stage CD71+ erythroid cells [27]. In the present analysis of gene expression in the NanoString nCounter Sprint system, a difference in gene expression was shown between natural bone marrow CD71+ erythroid cells and cultured (induced) CD71+ erythroid cells. These data indicate that: CD71+ erythroid cells cultured *in vitro* were more mature than erythroblasts of bone marrow origin, since they have a significantly lower expression of the ITGA4, TFRC, and CD36 genes [28] program of homeostatic regulation of stationary erythropoiesis. The balance between production and destruction of red blood cells is tightly regulated to maintain the optimal concentration of red blood cells in the blood. Too few erythrocytes are not able to provide sufficient oxygen delivery to the

tissues, and too many erythrocytes lead to an increase in blood viscosity, impaired blood flow, which also reduces oxygen delivery. Induced erythroblasts do not have restrictive regulatory requirements, which allows them to quickly pass through the differentiation program [29]. Analysis of gene expression showed that natural and induced CD71+ erythroid cells differ significantly in the processes of activation of the mechanisms of the innate and adaptive immune systems, activation of T- and B-cell signaling, activation and apoptosis of lymphocytes, but have a similar effect on the functional differentiation of lymphocytes.

Previously, it was shown that bone marrow erythroid cells are able to produce a wide range of cytokines that can regulate various stages of hemo- and immunopoiesis [30–35]. In addition, the same work revealed differences in the production of some cytokines between natural bone marrow erythroblasts and cells obtained de novo from erythroid colonies [36]. In the current study, multiplex analysis showed that both types of CD71+ erythroid cells secrete numerous cytokines with different immunoregulatory and inflammatory properties, but induced CD71+ erythroid cells differ in the production of immunoregulatory molecules from natural adult bone marrow CD71+ erythroid cells. The overall cytokine profiles are similar between the two types of CD71+ erythroid cells, but the production of interleukin, chemokines, and growth factors differs significantly between the two populations. The most significant differences between the two types of erythroblasts are presented for IL-1beta, IL-13, RANTES and IL-12 (p40): their secretion by natural bone marrow CD71+ erythroid cells is more than 10 times higher than the secretion by induced CD71+ erythroid cells.

Most of the cytokines in the culture supernatant of CD71+ erythroid cells can be attributed to mediators that provide chemotaxis and migration of resting memory cells, activated CD4 + and CD8+ T cells, natural killer and dendritic cells, growth of lymphoid cells, progenitor T cells and mature T cells, as well as factors contributing to both the activation of T cells and the suppression of inflammatory reactions [37, 38]. T-cells are present in the bone marrow and make up approximately 3–8% of the total number of nucleated cells, and these T-cells usually show a lower CD4/CD8 ratio compared to blood T cells [39]. The bone marrow is a "reservoir" of memory T-cells as it contains central memory T-cells and effector memory T-cells [40, 41]. Most T-lymphocytes are circulating cells, and their localization in the bone marrow allows them to remain active [42]. In addition, T cells regulate normal physiological processes in the bone marrow, such as normal hematopoiesis and bone tissue homeostasis [43, 44]. Thus, it can be assumed that bone marrow CD71+ erythroid cells actively produce a spectrum of cytokines necessary for the functioning and survival of T-cells. According to our data, induced CD71+ erythroid produce less of these cytokines, possibly due to the lack of appropriate hematopoietic niches and microenvironment. On the other hand, induced CD71+ erythroid cells actively produce such cytokines as SCF, which promotes the proliferation of myeloid and lymphoid hematopoietic precursor cells in bone marrow cultures [45], as well as IL-3 and IL-5, which stimulate the formation of colonies of megakaryocytes, neutrophils, and macrophages from bone marrow cultures and growth of immature hematopoietic progenitors of BFU-E [46]. CD71+ erythroid cells secrete large amounts of G-CSF, which stimulates hematopoietic progenitor cells and promotes the formation and mobilization of bone marrow neutrophils, as well as the survival and chemotaxis of mature neutrophils [47]. Based on these data, it can be assumed that CD71+ erythroid cells, with the help of soluble factors, are capable of autocrine regulation not only of erythropoiesis, but also of the activity of other hematopoietic cells. Cytokines help create a microenvironment that promotes both self-renewal and recruitment of various cell populations, while the observed suppressive properties of CD71+ erythroid cells induced by us are mainly due to the observed low production of factors that stimulate the pro-inflammatory immune response [48, 49].

## Conclusion

Thus, differentiation of CD34+ cells in vitro induces an CD71+ erythroid cells phenotype with immunoregulatory activity different from that of natural bone marrow CD71+ erythroid cells. This view is supported by the agreement between our multiplex analysis and mRNA profiling results regarding differences in properties between these two groups of cells. Our findings are relevant and important, since CD71+ erythroid cells can have both a positive regulatory effect (in inflammatory and autoimmune diseases, pregnancy, and the neonatal period) and a negative regulatory effect (in cancer, infectious diseases, and anemia) on the mechanisms of the immune system [15]. Differentiation of erythrocytes in vitro is an important clinically significant process for modeling the study of the development of erythrocytes in normal and pathological hematopoiesis, for the development of therapeutic protocols aimed at the treatment of diseases associated with erythrocytes, such as sickle cell anemia, thalassemia, various anemias [1, 50, 51]. In addition to immunoregulatory properties, *in vitro* erythropoiesis is a good model for studying translational control in cell differentiation and physiology [52], antibody screening tools, disease models or developmental research tools [53]. Thus, when cultivating erythroblasts for clinical experimental studies, it is necessary to take into account their pronounced immunoregulatory activity.

## Author Contributions

**Conceptualization:** Olga Yu. Perik-Zavodskaya, Sergey V. Sennikov.

**Data curation:** Roman Yu Perik-Zavodskii, Olga Yu. Perik-Zavodskaya, Sergey V. Sennikov.

**Formal analysis:** Roman Yu Perik-Zavodskii, Olga Yu. Perik-Zavodskaya.

**Funding acquisition:** Sergey V. Sennikov.

**Investigation:** Julia A. Shevchenko, Kirill V. Nazarov, Yulia G. Philippova, Alaa Alsalloum.

**Methodology:** Julia A. Shevchenko, Kirill V. Nazarov, Vera V. Denisova, Yulia G. Philippova, Alaa Alsalloum.

**Project administration:** Sergey V. Sennikov.

**Resources:** Vera V. Denisova.

**Software:** Roman Yu Perik-Zavodskii, Olga Yu. Perik-Zavodskaya.

**Supervision:** Roman Yu Perik-Zavodskii, Vera V. Denisova.

**Validation:** Roman Yu Perik-Zavodskii, Vera V. Denisova, Olga Yu. Perik-Zavodskaya, Yulia G. Philippova, Alaa Alsalloum.

**Visualization:** Kirill V. Nazarov, Olga Yu. Perik-Zavodskaya.

**Writing – original draft:** Julia A. Shevchenko, Yulia G. Philippova.

**Writing – review & editing:** Sergey V. Sennikov.

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
