## [Decision Letter · Decision Letter 0]

18 Apr 2023

PONE-D-23-08362Immunoregulatory properties of erythroid nucleated cells induced from CD34+ progenitors from bone marrow

PLOS ONE

Dear Dr. Sennikov,

Thank you for submitting your manuscript to PLOS ONE. After careful consideration, we feel that it has merit but does not fully meet PLOS ONE’s publication criteria as it currently stands. Therefore, we invite you to submit a revised version of the manuscript that addresses the points raised during the review process.

The comments are attached  for your consideration. The comments and suggestions are to improve the quality of your presentation according to the interests of researchers in the field.

We look forward to receiving your revised manuscript.

Kind regards,

Sriram Chitta, Ph D

Academic Editor

PLOS ONE

Journal Requirements:

   "Sennikov SV

     grant number No. 21-15-00087

     Russian Science Foundation

     https://rscf.ru/en/project/21-15-00087/

     NO"

Additional Editor Comments:

1. Revise protocol for cytokines analysis with details: In current manuscript , it is not clear whether authors took sample supernatants directly from cultures in conditioned medium or from cultured after differentiation in a separate medium. If latter, what is the medium used.  The methods used were not clear whether cells were washed and re-cultured, for cytokine analysis. If samples were taken directly from conditioned medium, wondering whether cytokines present in the conditioned medium were excluded. The controls for cytokine analysis were not clear.

Reviewers' comments:

Reviewer's Responses to Questions

**Comments to the Author**

1. Is the manuscript technically sound, and do the data support the conclusions?

Reviewer #1: Yes

Reviewer #2: Yes

2. Has the statistical analysis been performed appropriately and rigorously? 

Reviewer #1: I Don't Know

Reviewer #2: Yes

3. Have the authors made all data underlying the findings in their manuscript fully available?

Reviewer #1: Yes

Reviewer #2: No

4. Is the manuscript presented in an intelligible fashion and written in standard English?

Reviewer #1: No

Reviewer #2: Yes

5. Review Comments to the Author

Reviewer #1: The manuscript by Shevchenko et al. aims to characterize erythroid progenitors and further compare their immunological properties. They have proposed comparing bone marrow-derived versus induced erythroid progenitors using conditioning culture media.

Although the study subject is very interesting and could shed light on the biological properties of these cells, data are not well presented.

The major concern is the manuscript structure. Instead of discussing different sections in a more meaningful way, the authors are presenting their results in figure legends. The figure legends should come at the end of the manuscript.

The other major concern is that there is no evidence for the purity of isolated cells from the bone marrow and cell viability. My main concern is the lack of cell purity/viability. These two variable can fundamentally change the their results. Low cell purity/viability can fundamentally change the results.

I am hesitant that some of genes or cytokines such as IL-17, IL-13, IL-5 and etc might not be associated with these cells but the presence of other cells. In another word, cell contamination may skew their results. Considering that myeloid and erythroid cells have the same lineage, conditioning media may skew erythroid cells to myeloid instead of erythroid. Please see a recent paper in Cancer Cell by Long et al. 2022. PMID: 35594863.

Therefore, cell viability and purity can substantially alter the research outcomes and should be discussed. The best solution would be providing flow plots for the cell purity and viability.

If possible please include and discuss a couple of recent articles related to immunosuppressive role of CD71+ erythroid cells and their susceptibility to SARS-CoV-2 infection (Shahbaz S. et al. Stem cell reports 2021, PMID: 33979601, and Saito et al. Microbiology Spectrum 2022, PMID: 35943266).

Moreover, the authors should check the sex effect in their study as recently was reported that CD71+ erythroid cells are not only more abundant in females than males but also, they exert more immunosuppressive proprieties. (Mashhouri et al. Frontiers in Immunology, MID: 34367164 ).

The quality of plots/figures need to be improved as it was impossible to read the genes from their RNASeq.

Reviewer #2: In their article, Shevchenko et al. investigated whether erythroid cells derived from CD34+ progenitors from bone marrow have immunoregulatory properties. Authors found that erythroblasts derived from CD34+ cells carry the main markers of erythroid cells, but differ from bone marrow erythroblasts. The immunoregulatory role of CD71+ erythroid cells (CECs) is currently extensively investigated. The article by Shevchenko is therefore important, even it it rather a research letter. The novelty is rather low since their finding are consistent with the literature. However, up to now it was demonstrated that CEC generated ex vivo from Peripheral blood mononucelar cells have immunoregulatory properties. There is no report that CD34+ cells from BM differentiated into CEC have the same properties. Moreover, transcriptional analysis comparing ex vivo generated CEC and bone marrow CEC are interesting.

Introduction is well written and provides background of the study. It should be added that recent study demonstrated that potent suppression of T-cells is a general feature of CECs regardless of the trigger of their expansion (Grzywa et al Comm Biol 2021). Materials and methods are described comprehensively. Results are described clearly. However, the quality of the figures has to be improved. Discussion is interesting and well-written.

Major points

1. Authors should change the nomenclature of investigated cells from "erythroid nucleated cells" or "erythroblast" to "CECs" - "CD71+ erythroid cells". Using the term CD71+ erythroid cells is scientifically more acceptable which is covered by the literature.

2. What was the purity of isolated CD34+ cells? What was the purity of isolated CD71+ cells? Authors should provide this data.

Minor points

1. Line 46 "promote infiltration" - infiltration of what? Immune cells? Tumor-promoting cells? It has to be clafiried.

2. Line 47 "erythroblasts (...) lead to hypoxia, anemia, and coagulopathy due to expansion in the peripheral blood during SARS-CoV-2 infection". This sentence is not true. Erythroblast do not lead to the hypoxia and anemia. Their expansion is A CONSEQUENCE of hypoxia and anemia. It has to be corrected.

3. Line 200 - "Erythroblasts are a heterogeneous population, but they provide their main and immune cells with cells of all stages" - this sentence is not true. Please see Grzywa et al. Comm Biol 2021.

4. Methods - change "105 cells/well" to "10 5(superscript) cells/well"

6. PLOS authors have the option to publish the peer review history of their article (what does this mean?). If published, this will include your full peer review and any attached files.

Reviewer #1: **Yes: **Shokrollah Elahi

Reviewer #2: No

---

## [Author Response · Author response to Decision Letter 0]

29 May 2023

Dear editor! We have tried to implement all your recommendations.

We designed the manuscript using PLOS ONE style templates. We have corrected many errors in the text of the manuscript.

 "Sennikov SV

 grant number No. 21-15-00087

 Russian Science Foundation

https://rscf.ru/en/project/21-15-00087/

 NO"

We have added the necessary information. Dear editor, we would like to change the name of the study sponsor to "The study was carried out at the expense of State Assignment No. 122011800108-0". We did the study at our institution and it is very important for us to indicate this in the article. This order was received after the submission of the article, so we did not enter such information. Please give us an answer if possible.

Our data contains no ethical or legal restrictions on the exchange of a set of anonymized data. We had difficulty with this question, since we had no experience with placing data in repositories. In the ZENODO repository, Yulia Shevchenko posted minimal data on cytometric analysis of cells, cytokine production, and the results of a mixed leukocyte reaction. DOI:10.5281/zenodo.7865730. Roman Perik-Zavodskii posted his RNA sequencing data in another database: Gene Expression Omnibus - escrow code GSE231654. 

Additional Editor Comments:

Revise protocol for cytokines analysis with details: In current manuscript , it is not clear whether authors took sample supernatants directly from cultures in conditioned medium or from cultured after differentiation in a separate medium. If latter, what is the medium used. The methods used were not clear whether cells were washed and re-cultured, for cytokine analysis. If samples were taken directly from conditioned medium, wondering whether cytokines present in the conditioned medium were excluded. The controls for cytokine analysis were not clear.

Natural CD71+ erythroid cells from bone marrow. We isolated CD71+ erythroid cells from bone marrow using magnetic sorting. The resulting cells were cultured in X-VIVO 15 medium for 72 hours. We collected the supernatant of this culture and used it for cytokine analysis.

We grew induced CD71+ erythroid cells from bone marrow CD34+ cells. The cultivation process took place in three stages: expansion stage, stage of cell differentiation, stage of cell maturation. After each stage of cultivation, the cells were harvested, washed twice in Dulbecco's solution, and then cultured in a new portion of the culture medium. Thus, the cytokines that we used to differentiate erythroid cells did not get into the subsequent culture. We analyzed the supernatant of erythroid cells after the third phase of cultivation. At this stage, only erythropoietin, which is not included in the cytokine assay kit, was added to the cells for maturation. Thus, all the cytokines that we analyzed are secretion products of erythroid cells. Controls for the cytokine assay are present in the cytokine assay kit itself. The kit contains a control sample. An additional control is a blank, which is only a medium for cultivation.

Reviewer #1: The manuscript by Shevchenko et al. aims to characterize erythroid progenitors and further compare their immunological properties. They have proposed comparing bone marrow-derived versus induced erythroid progenitors using conditioning culture media. Although the study subject is very interesting and could shed light on the biological properties of these cells, data are not well presented.

Dear Doctor Elahi, we are honored that it was you who reviewed our work. We study your work with great interest and share this information with students and young doctors. A few weeks ago, we gave a lecture to obstetricians and gynecologists in which we discussed the role of erythroid cells in the mechanisms of fetomaternal tolerance. The young doctors were very interested in your work.

The major concern is the manuscript structure. Instead of discussing different sections in a more meaningful way, the authors are presenting their results in figure legends. The figure legends should come at the end of the manuscript.

We designed the manuscript in accordance with PLOS ONE style templates. In accordance with these recommendations, figure captions are also placed in the body of the article. We have created a Supporting information section with captions for the pictures. As I understand it, your remark refers to Figure 6. Indeed, this figure is accompanied by a very large text. We moved it to the main text and improved the discussion of the results.

The other major concern is that there is no evidence for the purity of isolated cells from the bone marrow and cell viability. My main concern is the lack of cell purity/viability. These two variable can fundamentally change the their results. Low cell purity/viability can fundamentally change the results.

We evaluated the purity and viability of the isolated cells. We considered these experiments to be routine, which each researcher conducts to control the quality of his work. I will present these data here, but if necessary, we will transfer them to the body of the manuscript. We examined the bone marrow almost immediately after the aspiration biopsy. No more than 30 minutes elapsed between obtaining the bone marrow and the process of cell isolation. The bone marrow was not subjected to conservation, which made it possible to keep the cells as viable as possible. The viability of isolated CD71+ erythroid cells was assessed visually by trypan blue staining on a Countess instrument. Cell viability was 95-99%. Next, we assessed cell viability using 7AAD and the purity of the isolated erythroid cells using anti-CD71 and anti-CD235 a antibodies during cytometric analysis. The data on the purity of the isolated cells are presented in Figure 1, but I will duplicate them here.

Dot-plot for natural bone marrow CD71+ erythroid cells

Histogram of the distribution of bone marrow CD71+ erythroid cells by 7AAD

Histogram of the distribution of live bone marrow CD71+ erythroid cells by marker CD 45

Histogram of the distribution of live bone marrow CD 45–CD71+ erythroid cells by marker CD 71

Histogram of the distribution of live bone marrow CD 45–CD71+ erythroid cells by marker CD 235a

I am hesitant that some of genes or cytokines such as IL-17, IL-13, IL-5 and etc might not be associated with these cells but the presence of other cells. In another word, cell contamination may skew their results. Considering that myeloid and erythroid cells have the same lineage, conditioning media may skew erythroid cells to myeloid instead of erythroid. Please see a recent paper in Cancer Cell by Long et al. 2022. PMID: 35594863.

Erythroid cells have not yet been fully studied; there is still much unknown in their history and functional properties. It is possible that our article will be the first mention of the production of IL-17, IL-13, IL-5 by erythroid cells. Yes, erythroid and myeloid cells have a common origin, but CD71 expression occurs only at the proerythroblast stage [Grzywa TM, Nowis D, Golab J. The role of CD71+ erythroid cells in the regulation of the immune response. Pharmacol Ther. 2021 Dec;228:107927. doi: 10.1016/j.pharmthera.2021.107927. Epub 2021 Jun 24. Erratum in: Pharmacol Ther. 2021 Nov 22;:108034. PMID: 34171326.], so the probability of contamination of CD71 cells with myeloid cells is negligible. The article you recommended to study, unfortunately, is not available in the full version, even using the Sci-hub service, so at the moment I have no way to compare our data with the data of the authors of that article (Cancer Cell by Long et al. 2022. PMID : 35594863.). In our work, IL-5 production is typical only for induced CD71+ erythroid cells; bone marrow erythroid cells do not produce this cytokine. It is possible that this is due precisely to differentiation outside the usual hematopoietic niche. IL-13, in contrast, is secreted only by CD71+ erythroid cells in the bone marrow. IL-13 is a structural and functional analog of IL-4 (Iwaszko M, Biały S, Bogunia-Kubik K. Significance of Interleukin (IL)-4 and IL-13 in Inflammatory Arthritis. Cells. 2021 Nov 3;10(11): 3000. doi: 10.3390/cells10113000. PMID: 34831223; PMCID: PMC8616130.). We have previously shown that bone marrow erythroid cells secrete IL-4 (Sennikov SV, Injelevskaya TV, Krysov SV, Silkov AN, Kovinev IB, Dyachkova NJ, Zenkov AN, Loseva MI, Kozlov VA. Production of hemo- and immunoregulatory cytokines by erythroblast antigen+ and glycophorin A+ cells from human bone marrow BMC Cell Biol 2004 Oct 18;5(1):39 doi: 10.1186/1471-2121-5-39 PMID: 15488155; liver secrete IL-4 (Sennikov SV, Krysov SV, Injelevskaya TV, Silkov AN, Kozlov VA. Production of cytokines by immature erythroid cells derived from human embryonic liver. Eur Cytokine Netw. 2001 Apr-Jun;12(2):274- 9. PMID: 11399516.), shows IL-4 gene expression in erythroid cells of mice (Sennikov SV, Eremina LV, Samarin DM, Avdeev IV, Kozlov VA. Cytokine gene expression in erythroid cells. Eur Cytokine Netw. 1996 Dec;7(4 ):771-4.PMID: 9010680.). Thus, we think that IL-13 is almost always present together with IL-4, it just attracted less interest from researchers in the past. One of the important functions of IL-17 is the induction of chemokines such as CXCL1, CXCL2 and CXCL8. We have previously shown that erythroid cells express chemokine receptor genes (Perik-Zavodskii R, Perik-Zavodskaia O, Shevchenko J, Denisova V, Alrhmoun S, Volynets M, Tereshchenko V, Zaitsev K, Sennikov S. Immune Transcriptome Study of Human Nucleated Erythroid Cells from Different Tissues by Single-Cell RNA-Sequencing Cells 2022 Nov 9;11(22):3537 doi: 10.3390/cells11223537 PMID: 36428967; PMCID: PMC9688070). Thus, we believe that IL-17 can be produced by erythroid cells to regulate chemokine expression.

Therefore, cell viability and purity can substantially alter the research outcomes and should be discussed. The best solution would be providing flow plots for the cell purity and viability.

Yes, we have made graphs that show the high purity and viability of erythroid cells. 

If possible please include and discuss a couple of recent articles related to immunosuppressive role of CD71+ erythroid cells and their susceptibility to SARS-CoV-2 infection (Shahbaz S. et al. Stem cell reports 2021, PMID: 33979601, and Saito et al. Microbiology Spectrum 2022, PMID: 35943266).

We thank you for the information about these excellent articles. We have cited them in the discussion. They confirm the theory that DM 45+ erythroblasts are not a research error, but are worthy elements of extramedullary erythropoiesis and have immunosuppressive properties.

Moreover, the authors should check the sex effect in their study as recently was reported that CD71+ erythroid cells are not only more abundant in females than males but also, they exert more immunosuppressive proprieties. (Mashhouri et al. Frontiers in Immunology, MID: 34367164 ).

We did not see any qualitative differences between men and women. Naturally, we would pay attention to this and use this fact in further research.

The quality of plots/figures need to be improved as it was impossible to read the genes from their RNASeq.

We have replaced this picture.

Reviewer #2: In their article, Shevchenko et al. investigated whether erythroid cells derived from CD34+ progenitors from bone marrow have immunoregulatory properties. Authors found that erythroblasts derived from CD34+ cells carry the main markers of erythroid cells, but differ from bone marrow erythroblasts. The immunoregulatory role of CD71+ erythroid cells (CECs) is currently extensively investigated. The article by Shevchenko is therefore important, even it it rather a research letter. The novelty is rather low since their finding are consistent with the literature. However, up to now it was demonstrated that CEC generated ex vivo from Peripheral blood mononucelar cells have immunoregulatory properties. There is no report that CD34+ cells from BM differentiated into CEC have the same properties. Moreover, transcriptional analysis comparing ex vivo generated CEC and bone marrow CEC are interesting.

Introduction is well written and provides background of the study. It should be added that recent study demonstrated that potent suppression of T-cells is a general feature of CECs regardless of the trigger of their expansion (Grzywa et al Comm Biol 2021). Materials and methods are described comprehensively. Results are described clearly. However, the quality of the figures has to be improved. Discussion is interesting and well-written.

Dear reviewer! I thank you for your appreciation of our work. Currently, the direction of the study of erythroid cells is actively developing, and we consider each study on this topic relevant and important.

Major points 

1. Authors should change the nomenclature of investigated cells from "erythroid nucleated cells" or "erythroblast" to "CECs" - "CD71+ erythroid cells". Using the term CD71+ erythroid cells is scientifically more acceptable which is covered by the literature.

We have replaced the terms "erythroid nucleated cells" or "erythroblast" with "CD71+ erythroid cells". In some places, we left the term "erythroblast" as more logical, for example, in the description of the stages of differentiation of CD71+ erythroid cells.

2. What was the purity of isolated CD34+ cells? What was the purity of isolated CD71+ cells? Authors should provide this data.

We evaluated the purity and viability of the isolated cells. We considered these experiments to be routine, which each researcher conducts to control the quality of his work. I will present these data here, but if necessary, we will transfer them to the body of the manuscript. We examined the bone marrow almost immediately after the aspiration biopsy. No more than 30 minutes elapsed between obtaining the bone marrow and the process of cell isolation. The bone marrow was not subjected to conservation, which made it possible to keep the cells as viable as possible. The viability of isolated CD71+ erythroid cells was assessed visually by trypan blue staining on a Countess instrument. Cell viability was 95-99%. Next, we assessed cell viability using 7AAD and the purity of the isolated erythroid cells using anti-CD71 and anti-CD235 a antibodies during cytometric analysis. The data on the purity of the isolated cells are presented in Figure 1, but I will duplicate them here.

Dot-plot for natural bone marrow CD71+ erythroid cells

Histogram of the distribution of bone marrow CD71+ erythroid cells by 7AAD

Histogram of the distribution of live bone marrow CD71+ erythroid cells by marker CD 45

Histogram of the distribution of live bone marrow CD 45–CD71+ erythroid cells by marker CD 71

Histogram of the distribution of live bone marrow CD 45–CD71+ erythroid cells by marker CD 235a

Dot-plot for natural bone marrow CD34+ cells after magnetic sorting 

Dot-plot of the distribution l bone marrow CD34+ cells after magnetic sorting by marker CD 34

Minor points 

1. Line 46 "promote infiltration" - infiltration of what? Immune cells? Tumor-promoting cells? It has to be clafiried.

We have replaced "promote infiltration" with "infiltrate the tumor and contribute to its progression". We wanted to say that erythroid cells infiltrate the tumor and contribute to its progression.

2. Line 47 "erythroblasts (...) lead to hypoxia, anemia, and coagulopathy due to expansion in the peripheral blood during SARS-CoV-2 infection". This sentence is not true. Erythroblast do not lead to the hypoxia and anemia. Their expansion is A CONSEQUENCE of hypoxia and anemia. It has to be corrected.

We have replaced "erythroblasts (...) lead to hypoxia, anemia, and coagulopathy due to expansion in the peripheral blood during SARS-CoV-2 infection" with “accumulate in the peripheral blood during SARS-CoV-2 infection, leading to hypoxia, anemia, and coagulopathy in moderate to severe infection”

3. Line 200 - "Erythroblasts are a heterogeneous population, but they provide their main and immune cells with cells of all stages" - this sentence is not true. Please see Grzywa et al. Comm Biol 2021.

Dear reviewer, it is possible that I did not accurately convey my idea. I changed this phrase to another "CD71+ erythroid cells are a heterogeneous population with several stages of differentiation. The cells of each stage realize their unique functions, which are difficult to separate at the level of the whole organism".

4. Methods - change "105 cells/well" to "10 5(superscript) cells/well"

We checked the text and corrected this error. It is possible that this error occurred due to text formatting.

Dear editor, we have placed our data in the Zenodo repository and received a link and DOI. After your remark about availability, we have checked our data. The data is loaded, but we realized that we need to access it in a different way. We provide you with the link https://zenodo.org/record/7865730. We have checked that the data is available on this link.

---

## [Editor Report · Decision Letter 1]

14 Jun 2023

Immunoregulatory properties of erythroid nucleated cells induced from CD34+ progenitors from bone marrow

PONE-D-23-08362R1

Dear Dr. Sergey  Sennikov

We’re pleased to inform you that your manuscript has been judged scientifically suitable for publication and will be formally accepted for publication once it meets all outstanding technical requirements.

Kind regards,

Sriram Chitta, Ph D

Academic Editor

PLOS ONE

Additional Editor Comments (optional):

Dear Dr Sennikov

I appreciate your efforts to answer all comments offered by reviewers and improve the manuscript accordingly. After carefully review the revised MS, I am recommending your work in PONE.

Sriram Chitta
---

## [Editor Report · Acceptance letter]

21 Jun 2023

PONE-D-23-08362R1 

Immunoregulatory properties of erythroid nucleated cells induced from CD34+ progenitors from bone marrow 

Dear Dr. Sennikov:

I'm pleased to inform you that your manuscript has been deemed suitable for publication in PLOS ONE. Congratulations! Your manuscript is now with our production department. 

Kind regards, 

on behalf of

Dr. Sriram Chitta 

Academic Editor

PLOS ONE